# An intelligent optimization method for highway route selection based on comprehensive weight and TOPSIS

Changjiang Liu[1,2]*, Qiuping Wang[1], Zhen Cao[1]

**1** School of Civil Engineering, Xi'an University of Architecture & Technology, Xi'an, China, **2** Shaanxi Province Transport Planning Design and Research Institute, Xian, China

* lcj2012@foxmail.com

## Abstract

In order to accurately analyze and evaluate multi-index and multi-route traffic schemes for comparison and selection, we introduce herein a comprehensive weight and an intelligent selection algorithm for traffic scheme optimization to improve upon the shortcomings of common qualitative and quantitative analysis methods. Firstly, we establish an evaluation index system of transportation by traffic scheme considering the factors of technology, ecological environment, social environment, and economy, based on the whole life cycle. Secondly, the comprehensive weight based on subjective and objective factors is constructed. Finally, we establish an optimization method for transportation schemes by using the comprehensive weight and Technique for Order Preference by Similarity to an Ideal Solution (TOPSIS) model. The results show that the evaluation index system based on the whole life cycle is more comprehensive and accurate. The comprehensive weight vector avoids the defects of single weight methods and makes full use of subjective data and expert opinions. The comprehensive weight vector is introduced into the decision-maker's preference coefficient, so that analysts can determine the scheme according to the subjective and objective information and to the required accuracy. This method uses a large number of evaluation groups to evaluate the scheme, and the evaluation results show greater objectivity and efficiency.

## Introduction

The selection of highway route schemes directly influences construction costs; construction and operation safety; the length of the construction period; and the costs relating to operation, management, and maintenance. In practical work, a trial-and-error method based on the work experience of designers and on expert evaluation is not time-consuming, but it can easily miss optimal schemes, and this process cannot be evaluated by experts and stakeholders with a large amount of data. For transportation schemes with complex geographical and societal considerations, it is more difficult to adapt this method to the requirements of scheme optimization. Therefore, a new intelligent optimization method is needed to quickly and comprehensively process data from a large number of evaluators and select the optimal scheme.

**Data Availability Statement:** Data is contained within the paper and its Supporting information files.

**Funding:** The author(s) received no specific funding for this work.

**Competing interests:** NO authors have competing interests.

Research on highway route schemes has been a topic of considerable interest in the field of highway design. Scholars have carried out extensive research in this field, and many meaningful results have been obtained. Gomes [1] studied the multi-criteria ranking of urban transportation system alternatives. Xiong et al. [2] studied the multi-criteria evaluation approach as applied to urban transportation projects. Kang et al. [3] integrated a genetic algorithm into a GIS platform and combined it with geographic information to optimize highway alignment. Kazemi et al. [4] introduced particle swarm optimization (PSO) and proposed a parallel PSO method to find the optimal solution to a highway alignment problem. Ismutulla et al. [5] applied the theory of the gray correlation degree and built an evaluation method for a highway route scheme based on the gray weighted correlation. Luo et al. [6] proposed a fuzzy comprehensive evaluation and selection method based on fuzzy mathematics. Mu et al. [7] introduced a multilevel fuzzy evaluation method to compare and select a highway route scheme based on multilevel fuzzy evaluation theory. Based on AHP and fuzzy comprehensive evaluation, Zhang et al. [8–10] put forward an optimization method for highway route schemes in the permafrost region of the Qinghai–Tibet Plateau. Sarı and Sen [11] proposed a design for a minimum-cost path algorithm for highway route selection.

Although the evaluation and selection methods [12–16] for highway route schemes can determine a proper route to a certain extent, their evaluation index systems focus on economic and technical indexes during construction and do not completely consider construction costs and environmental influences over the full life cycle. In addition, each subjective or objective weighting method is applied only to assign weights to indexes, which ignores the influences of objective factors, evaluators' opinions, and analyzers' preferences on indexes. As a result, the highway route scheme determined is not optimal [17, 18]. In addition, without the help of computers and intelligent technology, these methods cannot adapt to evaluation by a large group, and no feasible algorithm is applied to scheme optimization.

Many MCDM models can be used in optimal selection, such as RAFSI, MABAC, MAIRCA, VIKOR methods [19–22]. This method can make full use of the information of the original data, and its results can accurately reflect the gap between the evaluation schemes. Considering that Similarity to an Ideal Solution (TOPSIS) method is simpler, has no strict restrictions on data distribution and sample size, and is more suitable for internal sorting of sample data, this paper adopts TOPSIS method.On this basis, to make traffic optimization more efficient and scientific by means of big data, the existing research deficiencies and the objective needs of scheme selection should be considered as a whole. This paper proposes a solution that considers the full life cycle as a measurement state and builds an evaluation index system for a route scheme with comprehensive weight as an instrument to comprehensively select the weight for each index, introduces the Technique for Order Preference by TOPSIS model, and selects the optimal scheme from multiple schemes algorithmically. The AHP is effectively combined with the entropy weight method to determine the index weight, which avoids the defects of the single weighting method and takes into consideration the evaluator's opinions, the actual conditions of the highway, and the analyzer's preferences for setting the weights of the evaluation index. The selection of the route scheme is consistent with the calculation method for an ideal solution. The TOPSIS method was applied to evaluate a highway route scheme and determine the optimal route scheme based on the final grade.

## Materials and methods

### Evaluation index system for highway route schemes

**Selecting the evaluation index system.** When evaluating a highway route scheme, it is necessary to build the evaluation index system first. At present, the commonly used index

**Table 1. Evaluation index system for a highway route scheme.**

| Target layer | Evaluation layer | Operation layer | | Classification |
|---|---|---|---|---|
| **Evaluation index sytem for schemes** | Technical indexes *B1* | C11 | Route length | Quantitative- |
| | | C12 | Minimum curve of horizontal curve | Qualitative- |
| | | C13 | Operating speed | Quantitative |
| | | C14 | Average longitudinal slope | Quantitative- |
| | | C15 | Length of bridge | Quantitative- |
| | | C16 | Length of tunnel | Quantitative- |
| | | C17 | Coordination of average longitudinal slope | Qualitative+ |
| | | C18 | Highway capacity | Qualitative+ |
| | Ecological environment indexes *B2* | C21 | Engineering geology | Qualitative+ |
| | | C22 | Influences on sensitive environmental areas | Qualitative+ |
| | | C23 | Capacity to resist natural disasters during operation | Qualitative+ |
| | | C24 | Influences on mineral resources | Qualitative+ |
| | | C25 | Safety risks during construction | Qualitative+ |
| | Social environment indexes *B3* | C31 | Land acquisition | Quantitative- |
| | | C32 | Buildings to be demolished | Qualitative+ |
| | | C33 | Coordination with transport network in region | Qualitative+ |
| | | C34 | Coordination with planning for surrounding towns | Quantitative- |
| | Economic indexes *B4* | C41 | Construction cost | Quantitative- |
| | | C42 | Operation, management, and maintenance costs | Quantitative- |
| | | C43 | Operation costs of vehicles | Quantitative- |
| | | C44 | Construction period | Qualitative+ |
| | | C45 | Social and economic effect | Qualitative+ |

In a specific scheme, all or some of the indicators can be selected according to the characteristics of the project, and indicators can be added as required.

system generally uses the economic, technical, and environmental indicators during the construction period [11, 23–28]. It does not consider safety factors, and it lacks indicators of maintenance and management costs during the operation period; this cannot accurately and comprehensively reflect the scheme. In order to comprehensively consider the indicators of the scheme, we used the AHP to analyze factors influencing the scheme and built a full-life-cycle index system for highways that considers technology, the ecological environment, the social environment, and engineering economy (see Table 1).

Indexes can be divided into qualitative indexes and quantitative indexes based on their classification methods. According to trend, they can be divided into positive and negative categories; for a positive index (marked with "+"), a scheme will be more valuable when it is larger, and for a negative index (marked with "-"), a scheme will be more valuable when it is smaller.

**Quantitative indexes.** Quantitative indexes are analyzed according to their corresponding values. If there are $m$ evaluation schemes and $n$ quantitative indexes, and each scheme has a quantitative value $B_{ij}$ for each quantitative index (e.g., scheme length or construction costs), then

$$B_i^+ = \{b_{i1}, b_{i2}, \cdots, b_{in}\}, i = 1, 2, \cdots, m \tag{1}$$

For an index with a negative tendency, positive processing (similar to a tendency) [29] is required:

$$B_i^- = \{1/b_{i1}, 1/b_{i2}, \cdots, 1/b_{in}\}, i = 1, 2, \cdots, m \tag{2}$$

As an example, take three design schemes that were formulated for a certain highway with construction costs (CNY 100 million) of

$$B = \{1.2457, 1.3467, 1.2859\},$$

$$B_i^- = \{0.8028, 0.7526, 0.7777\}.$$

**Qualitative indexes.** Qualitative indexes are obtained from fuzzy classifications in the data. The fuzzy classifications have nine levels each: excellent, good, medium, bad, and poor, and large, general, and small. The evaluation scale used is the (1/9, 9) scale method.

**Weight vectors of the index system.** By getting the score of each index according to the rating method of the evaluators and normalizing the weight of each index [30], subjective randomness and preference can be inferred to a certain extent; this is called the subjective weight. Objective weight is calculated from quantitative indexes. To consider an evaluator's subjective perception of indexes and objective information among indexes, as well as an analyzer's preferences, this paper introduces a new method for determining weights, i.e., a comprehensive weight method.

**Calculating the subjective weight of an index.** The subjective weight is calculated via the analytic hierarchy process (AHP) [31], which is scored by the evaluators according to laws, regulations, experience, and interests. The steps are as follows.

Step 1: Create the weight judgment matrix.

A hierarchy model is established to compare the indexes of the criterion layer, and the relative importance of the indexes is obtained. The judgment matrix is constructed by the (1/9, 9) scale method. See Table 2 for the general form of the judgment matrix.

Step 2: Calculate the relative weight of each factor at each level.

According to the judgment matrix, the eigenvalues and the maximum eigenvalues of each factor are calculated by the square root method.

$$\begin{cases} M_i = \prod_{j=1}^{n} f_{ij}^k \\ \overline{\eta_i^k} = \sqrt[n]{M_i} \\ \eta_i^k = \overline{\eta_i^k} / \sum_{i=1}^{n} \overline{\eta_i^k} \\ \eta^k = \left( \eta_1^k, \eta_1^k, \cdots, \eta_n^k \right)^T \\ \lambda_{max}^k \frac{1}{n} \sum_{i,j=1}^{n} \left( \sum_{i,j=1}^{n} f_{ij}^k \eta_m^k \right) / \eta_i^k \end{cases} \quad (3)$$

Here,
$M_i$ is the row element product of the row in which the $i^{th}$ index of the $k^{th}$ evaluator in the judgment matrix is located;
$\eta_i^k$ is the relative weight of $\overline{\eta_i^k}$ after normalization;
$\eta^k$ is the eigenvector;
$\lambda_{max}^k$ is the maximum eigenvalue.

**Table 2. General form of the judgment matrix.**

| index | $F_1^{\,k}$ | . . . | $F_j^{\,k}$ | . . . | $F_n^{\,k}$ |
|---|---|---|---|---|---|
| $F_1^{\,k}$ | $f_{11}^{\,k}$ | . . . | $f_{1j}^{\,k}$ | . . . | $f_{1n}^{\,k}$ |
| . . . | . . . | . . . | . . . | . . . | . . . |
| $F_i^{\,k}$ | $f_{i1}^{\,k}$ | . . . | $f_{ij}^{\,k}$ | | $f_{in}^{\,k}$ |
| . . . | . . . | . . . | . . . | . . . | . . . |
| $F_n^{\,k}$ | $f_{n1}^{\,k}$ | | $f_{nj}^{\,k}$ | | $f_{nn}^{\,k}$ |

In Table 2, $h$ is the total number of evaluators; $k$ denotes the $k^{\text{th}}$ evaluator; $n$ is the total number of indicators in the criteria layer; $F_i^k$ and $F_j^k$ are the $i^{\text{th}}$ and $j^{\text{th}}$ indicators of the $k^{\text{th}}$ evaluator in the criteria layer, respectively; and $f_{ij}^{\,k}$ is the importance of indicator $F_i^{\,k}$ compared with indicator $F_j^k$. The value is determined according to the (1/9, 9) scale method by each evaluator.

Step 3: Calculate the consistency index.

To ensure that a weight is reasonable, it is necessary to check the consistency of each judgment matrix to determine whether it has satisfactory consistency. If it does not, the judgment matrix should be modified until it meets the consistency requirements. The formula for checking consistency is as follows:

$$\begin{cases} K = \dfrac{\lambda_{max-E}}{E-1} \\ G = \dfrac{K}{R} \end{cases} \tag{4}$$

where
$K$ is the consistency index;
$G$ is the random consistency ratio;
$E$ is the order of the judgment matrix;
$R$ is the random consistency index corresponding to the order of the judgment matrix.

Step 4: Find the subjective weight vector.

From the weight and the average social impact weight of each evaluation, the subjective weight can be calculated as follows:

$$\eta_j = \frac{1}{h}\sum_{k=1}^{h} \eta_j^k \quad \eta = (\eta_1, \eta_2, \cdots, \eta_n)^T \tag{5}$$

where $\sum_{j=1}^{n} \eta_j = 1$, $\eta_j \geq 0 (j = 1,2,\ldots,n)$.

**Calculating the objective weight of an index.** Information entropy is used for finding the objective weight. The data in this method are obtained from a numerical analysis of the highway route scheme evaluation index system.

Step 1: Generate the decision matrix. If there are $m$ evaluation schemes and $n$ evaluation indexes, then $d_{ij}$ ($i = 1, 2, \ldots,m; j = 1, 2, \ldots,n$), and the index weight matrix $D$ can be

expressed as

$$D = (d_{ij})_{m \times n} = \begin{bmatrix} d_{11} & d_{12} & \cdots & d_{1n} \\ d_{21} & d_{22} & \cdots & d_{2n} \\ \cdots & \cdots & \cdots & \cdots \\ d_{m1} & d_{m1} & \cdots & d_{mn} \end{bmatrix} \quad (6)$$

Step 2: Normalize the decision matrix. To eliminate the influences of each evaluation index on the evaluation of the route scheme due to the different dimensions of each evaluation index, normalization is required. In normalization, a decision matrix $D$ generates a standard matrix $V = (v_{ij})_{mn}$. The normalized value is found as follows:

$$v_{ij} = \left[ d_{ij} - min(d_j) \right] / \left[ max(d_j) - min(d_j) \right] \quad (7)$$

Step 3: Calculate the weight. If the feature weight of the $i^{th}$ evaluation object is $P_{ij}$ under the $j^{th}$ index, then

$$P_{ij} = v_{ij} / \sum_{i=1}^{m} v_{ij} \quad (8)$$

Step 4: Calculate the entropy $e_j$ of the $j^{th}$ index:

$$e_j = -\frac{1}{ln(m)} \sum_{i=1}^{m} P_{ij}(P_{ij}) \quad (9)$$

Step 5: Calculate the coefficient of difference $d_j$ for the $j^{th}$ index. For a certain index $d_j$, the smaller the difference $v_{ij}$ is, the larger $d_j$ will be. When the values of the $j^{th}$ indexes for each evaluated object are equal, then $e_j = e_{max}$ and $d_j$ will be

$$d_j = 1 i e_j \quad (10)$$

Step 6: Calculate the entropy weight of each index:

$$\mu_j = d_j / \sum_{k=1}^{n} d_k, j = 1, 2, \cdots, n \quad (11)$$

$$\mu = (\mu_1, \mu_2, \cdots, \mu_n)^T \quad (12)$$

where $\sum_{j=1}^{n} \mu_j = 1, \mu_j \geq 0$ ($j = 1, 2, \ldots, n$).

**Comprehensive weight.** To consider an evaluator's subjective perception of indexes and objective information among indexes, as well as an analyzer's preferences, we introduce a new method for determining weights, i.e., a comprehensive weight method.

If the comprehensive weight of each index is expressed as

$$\omega = (\omega_1, \omega_2, \cdots, \omega_n)^T \tag{13}$$

where $\sum_{j=1}^{m} \omega_l = 1$, $\omega_j \geq 0$ ($j = 1, 2, \ldots, m$), $z_{ij}$ is the evaluation matrix after standardization, then the comprehensive weighted score of a scheme is

$$f_i = \sum_{j=1}^{n} \omega_j z_{ij} i = 1, 2, \ldots, n \tag{14}$$

To both give consideration to subjective preference (for a subjective or objective weighting method) and make full use of the information provided by the subjective weighting method and the objective weighting method, and thereby achieve unity of the subjective and objective methods, the following optimization decision-making model is established:

$$\min F(\omega) = \sum_{i=1}^{m} \sum_{j=1}^{n} \left\{ \rho \left[ \left( \omega_j - \eta \right) z_{ij} \right]^2 + (1 - \rho) \left[ \left( \omega_j - \mu \right) z_{ij} \right]^2 \right\}$$

$$\text{s.t.} \begin{cases} \sum_{j=1}^{n} \omega_j = 1 \\ \\ \omega_j \geq 0, j = 1, 2, \cdots, n \end{cases} \tag{15}$$

where $0 \leq \rho \leq 1$ reflects the decision-maker's preference for subjective weight, named preference coefficient. The higher the value of $\rho$ is the decision maker prefer subjective weight.

Theorem 1 (S1 Appendix): If $\sum_{i=1}^{m} z_{ij}^2 > \mathbf{0}$ ($j = 1, 2, \ldots, n$), then the optimization formula (15) has a unique solution, which is

$$\omega = [\rho\eta_1 + (1-\rho)\mu_1, \rho\eta_2 + (1-\rho)\mu_2, \cdots, \rho\eta_n + (1-\rho)\mu_n]^T \tag{16}$$

**TOPSIS evaluation model.** Many methods can be used in optimal selection, such as TOPSIS, LBWA [32], FUCOM [33] or BWM models. However, considering that TOPSIS is simpler and more suitable for internal sorting of sample data, this paper adopts TOPSIS. TOPSIS [34, 35] refers to the technique for order preference by similarity to an ideal solution, the basic concept of which is to determine the optimal solution and the worst solution for a normalized original data matrix, then calculate the distance between the evaluated solution and the optimal solution and the worst solution, obtain the nearness degree between the evaluated solution and the optimal solution, and, on this basis, assess the advantages and disadvantages of each evaluated object. This method is widely used in multi-objective scheme selection, such as for highways, electric power, etc. [17, 36–39].

Step 1: If there are $n$ evaluation objects and $m$ evaluation indexes, then we can obtain an $m \times n$ initial judgment matrix $V$:

$$V = \begin{bmatrix} x_{11} & x_{12} & \cdots & x_{1n} \\ x_{21} & x_{21} & \cdots & x_{2n} \\ \vdots & \vdots & \vdots & \vdots \\ x_{i1} & \cdots & x_{ij} & \cdots \\ \vdots & \vdots & \vdots & \vdots \\ x_{m1} & x_{m1} & \cdots & x_{mn} \end{bmatrix} \tag{17}$$

Step 2: The dimension of each index may be different, so that decision matrix should be normalized:

$$
V' = \begin{bmatrix}
x'_{11} & x'_{1n} & \cdots & x'_{1n} \\
x'_{21} & x'_{22} & \cdots & x'_{2n} \\
\vdots & \vdots & \vdots & \vdots \\
x'_{i1} & \cdots & x'_{ij} & \cdots \\
\vdots & \vdots & \vdots & \vdots \\
x'_{m1} & x'_{m1} & \cdots & x'_{mn}
\end{bmatrix}
\tag{18}
$$

Where

$$
x'_{ij} = \frac{x_{ij}}{\sqrt{\sum_{i=1}^{n} x_{ij}^2}}, i = 1, 2, \cdots, m; j = 1, 2, \cdots, n
\tag{19}
$$

Step 3: Calculate the weight vector of each index from Eq (16) and generate a weighted judgment matrix.

$$
Z = V'W = \begin{bmatrix}
x'_{11} & x'_{1n} & \cdots & x'_{1n} \\
x'_{21} & x'_{22} & \cdots & x'_{2n} \\
\vdots & \vdots & \vdots & \vdots \\
x'_{i1} & \cdots & x'_{ij} & \cdots \\
\vdots & \vdots & \vdots & \vdots \\
x'_{m1} & x'_{m1} & \cdots & x'_{mn}
\end{bmatrix} \cdot \begin{bmatrix}
\omega_1 & 0 & \cdots & 0 \\
0 & \omega_2 & \cdots & 0 \\
\vdots & \vdots & \vdots & \vdots \\
0 & \cdots & \omega_i & \cdots \\
\vdots & \vdots & \vdots & \vdots \\
0 & 0 & \cdots & \omega_n
\end{bmatrix} = \begin{bmatrix}
z_{11} & z_{12} & \cdots & z_{1n} \\
z_{2n} & z_{22} & \cdots & z_{2n} \\
\vdots & \vdots & \vdots & \vdots \\
z_{i1} & \cdots & z_{ij} & \cdots \\
\vdots & \vdots & \vdots & \vdots \\
z_{m1} & z_{m2} & \cdots & z_{mn}
\end{bmatrix}
\tag{20}
$$

Step 4: Calculate the positive and negative ideal solutions of the evaluated targets according to the weighted judgment matrix.

The positive ideal solution is $z_j^+$:

$$
z_j^+ = \begin{cases}
max(z_{ij}), j \in J^+ \\
min(z_{ij}), j \in J^-
\end{cases}
\tag{21}
$$

The negative ideal solution is $z_j^-$:

$$
z_j^- = \begin{cases}
min(z_{ij}), j \in J^+ \\
max(z_{ij}), j \in J^-
\end{cases}
\tag{22}
$$

where

$j^+$ refers to the benefit index;

$j^-$ refers to the cost index.

Step 5: Calculate the Euclidean distance between each target value and the ideal value, $S_i^+, S_i^-$.

$$S_i^+ = \sqrt{\sum_{j=1}^{m}(f_{ij} - f_j^+)^2}, j = 1, 2, \cdots, n \tag{23}$$

$$S_i^- = \sqrt{\sum_{j=1}^{m}(f_{ij} - f_j^-)^2}, j = 1, 2, \cdots, n \tag{24}$$

Step 6: Calculate the relative degree of closeness of each target, $C_i^+$.

$$C_i^+ = \frac{S_i^-}{S_i^+ + S_i^-}, i = 1, 2, \ldots, m \,\&\, C_i^+ \in (0, 1) \tag{25}$$

Step 7: Sort the targets based on the relative degree of closeness and generate the decision criteria. When the $C_i^+$ value approaches 1, the evaluation object becomes closer to the positive ideal solution.

## Results

### Highway route scheme optimization algorithm

First, the indicator system of the highway route scheme can be constructed according to the characteristics of the project. All or some of the indicators in Table 1 can be selected, and some indicators can be added according to the characteristics of the project. The qualitative indexes and quantitative indexes in the index system are processed to obtain quantitative and positive index data and to build the initial judgment matrix.

Second, the subjective weight and objective weight are calculated.

Thirdly, the decision maker decided the preference coefficient and calculates the comprehensive weight.

Finally, the TOPSIS model is used to calculate the relative closeness of each scheme, and the optimal scheme is obtained (Fig 1).

### Route scheme selection automation

Highway route optimization algorithm involves a lot of calculation, especially when the number of experts and stakeholders is large. Therefore, it is necessary to use computer to calculate quickly to obtain the optimal scheme (Fig 2).

### Discussion and empirical research

The intelligent optimization method proposed in this paper establishes an index system based on the whole life cycle, which is comprehensive and avoids the adverse impact of incomplete index system on the results. This method takes into account both subjective weight and objective weight, and the two weights are calculated independently. At the same time, in order to consider the focus of decision-makers, preference coefficient is introduced. This method uses computer technology, can adapt to a large number of evaluation population, and can quickly

**Fig 1. The highway route scheme optimization algorithm.**

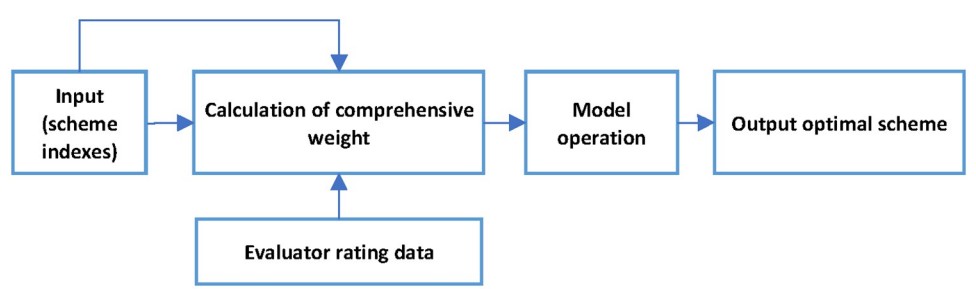

**Fig 2. Scheme optimization process.**

obtain the best scheme after evaluation. It can not only be used in highway scheme selection, but also can be used in other multi-objective scheme selection.

We took the route scheme for a highway in Tibet as an example for evaluation and comparison and obtained the optimal scheme through calculation. The length of the corridor from the Lalangqu to Zhagongqu is approximately 12 km, and it ascends the Xuegula mountain. Due to the complicated topography and geology of the area, a large average longitudinal slope, and a high proportion of bridges and tunnels, three schemes were formulated for comparison. Fig 3 shows the comparison and selection of highway route schemes.

Step 1: We obtained the index system data (the qualitative indexes and the quantitative indexes) of a multi-index highway route scheme, and then we built a judgment matrix. See Table 3 for the indexes.

Step 2: Twenty-three evaluators, comprising 7 experts and 16 stakeholders, scored the weight of each evaluation index. We calculated the comprehensive weight according to formulas (5), (12), and (16), as shown in Table 4.

According to the decision-maker's preference for qualitative and quantitative indexes, the preference coefficient ρ was 0.6.

Step 3: We normalized the judgment matrix and used the calculated comprehensive weight vector to build a weighted decision matrix. See Table 5 for the results.

Step 4: We calculated the relative degree of closeness between each scheme and the ideal solution. See Table 6 for the results.

In order to analyze the sensitivity of preference coefficient, we calculated the closeness under different preference coefficients, as shown in Fig 4.

It can be seen from Fig 4 that each scheme has a different closeness with different preference coefficient, and the optimal scheme is finally selected. *S1* with high closeness shall be preferred, when ρ ∈ (0,0.651). *S2* with high closeness shall be preferred, when ρ∈ (0.0651,1). It shows that the preference coefficient can affect and determine the conclusion of multi-objective scheme selection, that is an important sensitive factor in multi-objective scheme selection.

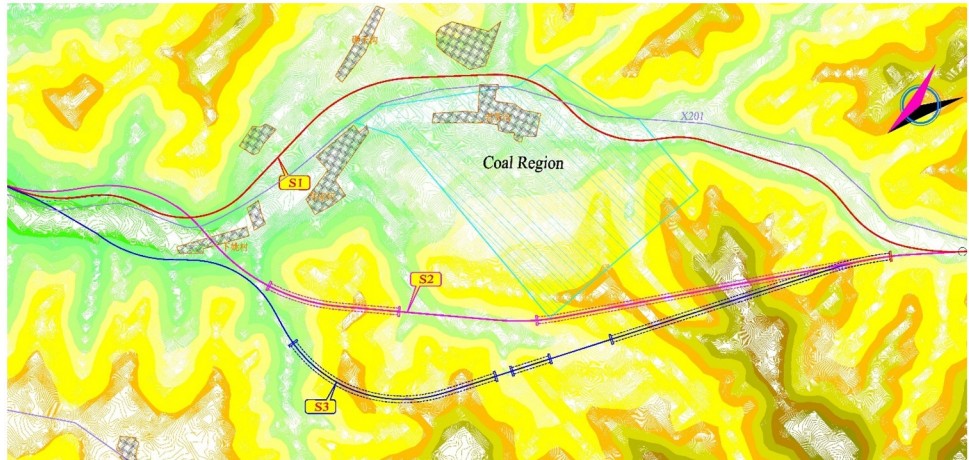

**Fig 3. Highway route plan.**

**Table 3. Evaluation indexes for the program.**

| Target layer | Evaluation layer | *S1* | *S2* | *S3* |
|---|---|---|---|---|
| Technical indexes | C11 | 11.21 | 10.64 | 10.80 |
| | C12 | 600/2 | 700/2 | 700/1 |
| | C13 | 98.4 | 102.4 | 103.7 |
| | C14 | 2.7/3.4 | 3.6/1.5 | 3.01/1.15 |
| | C15 | 2013/7 | 1767/8 | 1541/7 |
| | C16 | 0 | 4059.5/3 | 3532/2 |
| | C17 | Good | Excellent | Good |
| | C18 | Good | Excellent | Excellent |
| Ecological environment indexes | C21 | Qualified | Average | Average |
| | C22 | Qualified | Good | Good |
| | C23 | Qualified | Good | Good |
| | C24 | Good | Average | Poor |
| | C25 | Good | Poor | Fair |
| Social environment indexes | C31 | 833.4 | 742.8 | 753.5 |
| | C32 | 2470 | 1110 | 1470 |
| | C33 | Good | Average | Average |
| | C34 | Good | Average | Average |
| Economic indexes | C41 | 132073 | 141440 | 147796 |
| | C42 | 172 | 247 | 235 |
| | C43 | 2454 | 2330 | 2365 |
| | C44 | 24 | 38 | 34 |
| | C45 | Good | Excellent | Excellent |

The CS1 inversly decreased gradually along with the increase of ρ. It suggests that the main reason was that S1 has small scale, low cost and obvious advantages in overall objective indicators, but it covers a lot of land and has a long route, which leads to the low score of the evaluators. Along with ρ increase, $C_{S2}$ and $C_{S3}$ are increasing trends, but the increase of *S2* is greater, indicating that *S2* is right ρ More sensitive. The analysis shows that the main reason is that although *S2* tunnel is long, the route mileage is short, avoiding basic farmland, the cost and technical indicators are higher than S3, resulting in higher scores for the evaluators.

There are fewer human factors involved in the objective weight, and more evaluators in the objective weight. Considering from the perspective of economy, technology and stakeholders, the two weight schemes have their own advantages and disadvantages. Decision makers have a deeper understanding of the project and give preference factors based on the overall judgment of subjective and objective factors, considering their own interests, the environment during the construction period and the quality of evaluators. Decision makers can not easily judge the impact of preference factors on project conclusions, which not only considers the subjective and objective indicators and the decision-makers' evaluation of subjective and objective indicators, but also avoids the decision-makers from giving intuitive conclusions, which is conducive to ensuring the scientificity of scheme selection. If construction cost *C41*, operation, management, and maintenance costs *C42* and vehicle operation costs *C43* are ignored, then the relative degrees of closeness are in the order $C_{S3} > C_{S2} > C_{S1}$ (Table 7). Therefore, the construction of the indexes directly influences the evaluation results.

Through the above analysis and calculation, S3 is the optimal scheme. At the same time, the expert scoring method and the general comprehensive weight method are compared. This method is more in line with the expectations of the construction unit and experts.

**Table 4. Weight vectors.**

| B | C | η | μ | ω |
|---|---|---|---|---|
| B1- 0.2604 | C11 | 0.0537 | 0.0407 | 0.0485 |
| | C12 | 0.0244 | 0.0260 | 0.0251 |
| | C13 | 0.0197 | 0.0228 | 0.0209 |
| | C14 | 0.0337 | 0.0326 | 0.0332 |
| | C15 | 0.0332 | 0.0407 | 0.0362 |
| | C16 | 0.0429 | 0.0472 | 0.0446 |
| | C17 | 0.0227 | 0.0212 | 0.0221 |
| | C18 | 0.0301 | 0.0293 | 0.0298 |
| B2-0.2084 | C21 | 0.0951 | 0.0875 | 0.0921 |
| | C22 | 0.0217 | 0.0208 | 0.0214 |
| | C23 | 0.0326 | 0.0292 | 0.0312 |
| | C24 | 0.0218 | 0.0329 | 0.0263 |
| | C25 | 0.0372 | 0.0379 | 0.0375 |
| B3-0.1676 | C31 | 0.0475 | 0.0629 | 0.0536 |
| | C32 | 0.0623 | 0.0796 | 0.0692 |
| | C33 | 0.0241 | 0.0084 | 0.0178 |
| | C34 | 0.0337 | 0.0168 | 0.0269 |
| B4-0.3636 | C41 | 0.1717 | 0.1818 | 0.1757 |
| | C42 | 0.0553 | 0.0346 | 0.0470 |
| | C43 | 0.0427 | 0.0327 | 0.0387 |
| | C44 | 0.0313 | 0.0217 | 0.0275 |
| | C45 | 0.0626 | 0.0928 | 0.0747 |

## Conclusions

In this paper, we discussed the limitations of highway route selection. To address these limitations, we built an evaluation index system for highway route schemes based on the full life cycle of the route and proposed a new weight calculation method that uses the preference of decision-makers and comprehensively calculates the subjective weight and objective weight. We used a comprehensive weight vector and the TOPSIS model to build an evaluation system for a highway route scheme, and we verified the validity of this method of scheme evaluation using examples. The following conclusions can be drawn:

The construction of evaluation indexes directly influences the results of scheme evaluation. When subsequent operations, management, and maintenance indexes are included based on the full life cycle, the evaluation results are more accurate.

The comprehensive weight vector can overcome the defects of the single weight method; it not only makes use of objective data, but also fully considers experts' opinions. The comprehensive weight vector increases the preference coefficient ρ, which is essential different from existing other comprehensive weight methods. The preference coefficient ρ can be determined by the accuracy of subjective and objective information and preference. When the decision-maker obtains the subjective weight and objective weight, through comprehensive analysis of the project, a preference coefficient from the perspective of the interests of the decision-maker is proposed, which can better serve the decision-maker.

The evaluation results for the selected route scheme based on the comprehensive weight model with TOPSIS were basically consistent with on-site conditions; therefore, the method is feasible and valid for comprehensively evaluating a route scheme.

**Table 5. Comprehensive weight vector of the judgement matrix.**

| C | S1 | S2 | S3 |
|---|---|---|---|
| C11 | 0.94 | 1 | 0.98 |
| C12 | 0.92 | 0.94 | 1 |
| C13 | 0.94 | 0.98 | 1 |
| C14 | 1 | 0.75 | 0.89 |
| C15 | 0.76 | 0.87 | 1 |
| C16 | 1 | 0.35 | 0.41 |
| C17 | 0.91 | 1 | 0.92 |
| C18 | 0.96 | 1 | 0.99 |
| C21 | 0.46 | 0.92 | 1 |
| C22 | 0.64 | 1 | 0.95 |
| C23 | 1 | 0.44 | 0.35 |
| C24 | 0.46 | 0.92 | 1 |
| C25 | 1 | 0.12 | 0.26 |
| C31 | 0.89 | 1 | 0.98 |
| C32 | 0.44 | 1 | 0.75 |
| C33 | 1 | 0.76 | 0.64 |
| C34 | 1 | 0.64 | 0.71 |
| C41 | 1 | 0.84 | 0.81 |
| C42 | 1 | 0.69 | 0.73 |
| C43 | 0.94 | 1 | 0.98 |
| C44 | 1 | 0.42 | 0.69 |
| C45 | 0.86 | 1 | 0.94 |

**Table 6. Evaluation results of the relative degree of closeness.**

| PROG | S1 | S2 | S3 |
|---|---|---|---|
| Relative degree of closeness | 0.853 | 0.841 | 0.849 |

The degrees of closeness are in the order $C_{S1} > C_{S3} > C_{S2}$, so scheme *S1* is the best, followed by scheme *S3* and scheme *S2*. The evaluation results are completely consistent with on-site conditions.

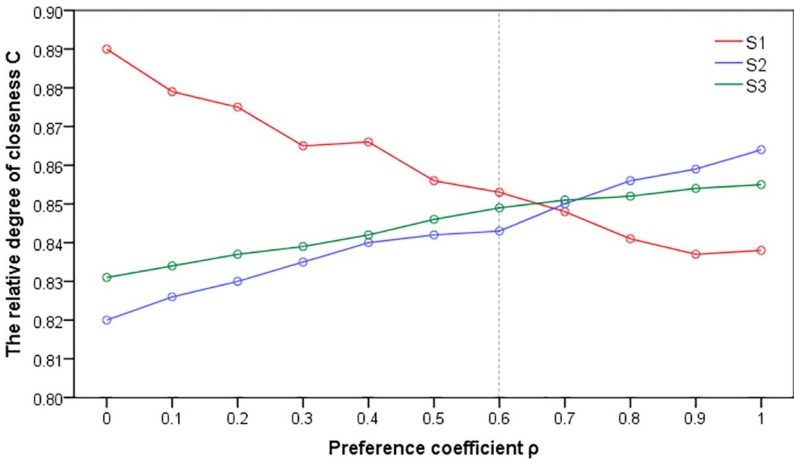

**Fig 4. Sensitivity analysis chart of the preference coefficient.**

**Table 7. Evaluation results of the relative degree of closeness.**

| PROG | S1 | S2 | S3 |
|---|---|---|---|
| Relative degree of closeness | 0.841 | 0.842 | 0.850 |

Intelligent computing can be used to form an evaluation of the scheme by a large number of evaluation groups (including experts, stakeholders, etc.), and the evaluation results are more objective. Intelligent Optimization Method can not only be used in highway scheme selection, but also can be used in other multi-objective scheme selection.

In future work, we will further study intelligent optimization methods of highway schemes. For example, we will study the method of selecting project indicators, differentiate the weight calculation methods of experts and stakeholders, and explore whether there is a better evaluation model that could replace TOPSIS.In addition, we will assess the usability of the approach for domain experts. Our ultimate objective is to use our scheme optimization approach for automated incident reporting, which can be made intelligent. The system only needs to input the index of the scheme and the score of the evaluation group, and the preference coefficient of the decision-maker can be used to automatically obtain the optimal scheme. To achieve this aim, we will use examples of potential selected schemes to evaluate the applicability of this method and to identify monitoring activities that may be useful in detecting or investigating the selection of these schemes.

## Supporting information

**S1 Appendix. Theorem 1: If $\sum_{i=1}^{m} z_{ij}^2 > 0$ ($j = 1, 2, \ldots, n$), then the optimization formula (15) has a unique solution.**
(DOCX)

**S1 Table. Index system data, scored the weight of 23 evaluators.**
(XLSX)

## Acknowledgments

The authors gratefully acknowledge Prof. Shaowei Yang and Prof. Yongyi Guo for useful discussions and consultations.

## Author Contributions

**Conceptualization:** Changjiang Liu.

**Data curation:** Changjiang Liu.

**Formal analysis:** Changjiang Liu, Zhen Cao.

**Funding acquisition:** Changjiang Liu.

**Investigation:** Changjiang Liu.

**Methodology:** Changjiang Liu.

**Project administration:** Changjiang Liu.

**Resources:** Changjiang Liu.

**Software:** Changjiang Liu.

**Supervision:** Changjiang Liu, Qiuping Wang, Zhen Cao.

**Validation:** Changjiang Liu, Qiuping Wang.

**Visualization:** Changjiang Liu.

**Writing – original draft:** Changjiang Liu.

**Writing – review & editing:** Changjiang Liu, Qiuping Wang, Zhen Cao.

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
