## [Decision Letter · Decision Letter 0]

6 May 2021

PONE-D-21-10954

An Intelligent Optimization Method for Highway Route Selection based on Comprehensive weight and TOPSIS

PLOS ONE

Dear Dr. Liu,

Thank you for submitting your manuscript to PLOS ONE. After careful consideration, we feel that it has merit but does not fully meet PLOS ONE’s publication criteria as it currently stands. Therefore, we invite you to submit a revised version of the manuscript that addresses the points raised during the review process.

We look forward to receiving your revised manuscript.

Kind regards,

Dragan Pamucar

Academic Editor

PLOS ONE

Journal Requirements:

2. Please amend your list of authors on the manuscript to ensure that each author is linked to an affiliation. Authors’ affiliations should reflect the institution where the work was done (if authors moved subsequently, you can also list the new affiliation stating “current affiliation:….” as necessary).

3. We note that [Figure 2] in your submission contain [map] images which may be copyrighted. All PLOS content is published under the Creative Commons Attribution License (CC BY 4.0), which means that the manuscript, images, and Supporting Information files will be freely available online, and any third party is permitted to access, download, copy, distribute, and use these materials in any way, even commercially, with proper attribution. For these reasons, we cannot publish previously copyrighted maps or satellite images created using proprietary data, such as Google software (Google Maps, Street View, and Earth). For more information, see our copyright guidelines: http://journals.plos.org/plosone/s/licenses-and-copyright.

3.1.    You may seek permission from the original copyright holder of Figure 2 to publish the content specifically under the CC BY 4.0 license. 

3.2.    If you are unable to obtain permission from the original copyright holder to publish these figures under the CC BY 4.0 license or if the copyright holder’s requirements are incompatible with the CC BY 4.0 license, please either i) remove the figure or ii) supply a replacement figure that complies with the CC BY 4.0 license. Please check copyright information on all replacement figures and update the figure caption with source information. If applicable, please specify in the figure caption text when a figure is similar but not identical to the original image and is therefore for illustrative purposes only.

4. Please upload a copy of Figure 3, to which you refer in your text on page 10. If the figure is no longer to be included as part of the submission please remove all reference to it within the text.

Reviewers' comments:

Reviewer's Responses to Questions

**Comments to the Author**

1. Is the manuscript technically sound, and do the data support the conclusions?

Reviewer #1: Partly

Reviewer #2: Yes

Reviewer #3: Yes

2. Has the statistical analysis been performed appropriately and rigorously? 

Reviewer #1: N/A

Reviewer #2: Yes

Reviewer #3: Yes

3. Have the authors made all data underlying the findings in their manuscript fully available?

Reviewer #1: Yes

Reviewer #2: Yes

Reviewer #3: Yes

4. Is the manuscript presented in an intelligible fashion and written in standard English?

Reviewer #1: Yes

Reviewer #2: Yes

Reviewer #3: Yes

5. Review Comments to the Author

Reviewer #1: Greetings,

The paper needs to be corrected to make it better.

The results of the research should be in the abstract. Materials and Methods should be corrected for example see the following papers:

Pamucar, D., & Ćirović, G. (2018). Vehicle route selection with an adaptive neuro fuzzy inference system in uncertainty conditions. Decision Making: Applications in Management and Engineering, 1 (1), 13-37.

Liu, F., Aiwu, G., Lukovac, V., & Vukic, M. (2018). A multicriteria model for the selection of the transport service provider: A single valued neutrosophic DEMATEL multicriteria model. Decision Making: Applications in Management and Engineering, 1 (2), 121-130.

Noureddine, M., & Ristic, M. (2019). Route planning for hazardous materials transportation: Multicriteria decision making approach. Decision Making: Applications in Management and Engineering, 2 (1), 66-85.

Blagojević, A., Vesković, S., Kasalica, S., Gojić, A., & Allamani, A. (2020). The application of the fuzzy AHP and DEA for measuring the efficiency of freight transport railway undertakings. Operational Research in Engineering Sciences: Theory and Applications, 3 (2), 1-23.

Kasalica, S., Obradović, M., Blagojević, A., Jeremić, D., & Vuković, M. (2020). Models for ranking railway crossings for safety improvement. Operational Research in Engineering Sciences: Theory and Applications, 3 (3), 84-100.

Algorithms cannot be results. Algorithms are a paper methodology. Correct this and better explain the methodology. The selection Discussion and empirical research is debatable. First is the Case study, then the results. There is no discussion. The results are incomplete. One result must be obtained with the TOPSIS method, and then a sensitivity analysis must be performed, in order to exclude some criteria, etc. It is necessary to make scenarios and then discuss all these results.

The conclusion should include the most important results, research limits and guidelines for future research.

Adjust the paper according to the guidelines.

Reviewer #2: First of all, the paper “An Intelligent Optimization Method for Highway Route Selection based on Comprehensive weight and TOPSIS” aims and scope match those of PLOS ONE, so the paper is adequate for this journal. This paper presents an application of TOPSIS for route selection selection. However based on my opinion it needs substantial improvements to be considered for publication in PLOS ONE. I would suggest a series of changes that in my opinion would improve the paper, in special for the reader.

- I suggest the authors to improve the introduction section. Authors should better highlight the objective of their work and to what extent it contributes to close a gap in the existing literature and/or practice. What is the innovative value of the contribution proposed by the authors?

- In introduction section authors should provide more information about existing MCDM models used in the field and their benefits/weaknesses.

- Why you have used TOPSIS method? Why not RAFSI, MABAC, MAIRCA, VIKOR methdos? This should be discussed. The authors need to discuss their contributions compared to those in related papers. The authors must clearly discuss the significance of the research problem in the first section.

- Add separate literature review section. You should provide more recent references published in last two-three years. Remove references published before 2017.

- The authors should explain why they insist in practically using the Comprehensive weight model. Why not LBWA, FUCOM or BWM models?

- You should provide step by step calculations for provided methodology, especially Comprehensive weight model. You should explain in detail this methodology.

- Explain in more details in the data used in the case study, the data for the testing, the criterion for the accuracy, and others to claim these points.

- Validation section is missing. How we can judge about these results? Comparisons with existing algorithms from the literature is missing.

- Discussion section is missing. How should we know about the quality of these solutions? Could you compare these results with some existing approaches in literature? The improvement must be discussed.

- The conclusion section seems to rush to the end. The authors will have to demonstrate the impact and insights of the research. The authors need to clearly provide several solid future research directions. Clearly state your unique research contributions in the conclusion section. Add limitations of the model. No bullets should be used in your conclusion section.

Reviewer #3: This paper constructed a comprehensive weight and an intelligent selection algorithm for traffic scheme optimization. Firstly, the authors established an evaluation index system considering the factors of the whole of life cycle. Secondly, the AHP method is used to determine the index weight, and the authors proposed the comprehensive weight algorithm based on subjective and objective factors. Finally, the TOPSIS method is used to rank the schemes. An application illustrates the effectiveness of the proposed method. However, there are some points the author needs to consider as follows:

1. It is suggested that the authors introduce the organization of this paper in the end of introduction.

2. Page 9, line 5, “and each scheme m has a quantitative value B_{ij}” should be “and each scheme has a quantitative value b_{ij} for each quantitative index”.

3. Page 10, line 16, “and f_i^k is the importance of indicator F_i^k compared with indicator F_j^k” should be “and f_{ij}^k is the importance of indicator F_i^k compared with indicator F_j^k”.

4. Page 11, formula (7), what does the symbols “x_{ij}, max(x_j), min(x_j)” stand for? Maybe they are “d_{ij}”, “max_j(d_{ij})” and “min_j(d_{ij})”.

5. Page 12, formula (14), what does the symbols “f_i” and “z_{ij}” stand for?

6. Page 12, formula (15), “where \\rho reflects the decision-maker’s preference for objective or subjective weight.” Does it reflect the preference of decision makers for objective or subjective weights here? The higher the value of \\rho is, does the decision maker prefer subjective weight or objective weight?

7. Page 12, in theorem 1, “formula (10)” should be “formula (15)”.

8. Page 13, in step 3, “formula (18)” should be “formula (16)”.

9. It is suggested that the authors compare the proposed comprehensive weight method with existing comprehensive weight methods.

10. It is suggested that the authors add a sensitivity analysis with variable \\rho.

Based on these comments, I suggest MAJOR REVISIONS before its acceptance.

6. PLOS authors have the option to publish the peer review history of their article (what does this mean?). If published, this will include your full peer review and any attached files.

Reviewer #1: No

Reviewer #2: No

Reviewer #3: No

---

## [Author Response · Author response to Decision Letter 0]

5 Nov 2021

Response to Journal Requirements

Point: 1. Please ensure that your manuscript meets PLOS ONE's style requirements, including those for file naming. The PLOS ONE style templates can be found at https://journals.plos.org/plosone/s/file?id=wjVg/PLOSOne_formatting_sample_main_body.pdf and（1）

Response: We revised according to requirements.

Point: 2. Please amend your list of authors on the manuscript to ensure that each author is linked to an affiliation. Authors’ affiliations should reflect the institution where the work was done (if authors moved subsequently, you can also list the new affiliation stating “current affiliation:….” as necessary).

Response: We have revised the Authors’ affiliations. We listed Professor Cao Zhen of Xi'an University of architecture and technology as the author and considered his contribution in Major Revision.

Point: 3. We note that [Figure 2] in your submission contain [map] images which may be copyrighted. All PLOS content is published under the Creative Commons Attribution License (CC BY 4.0), which means that the manuscript, images, and Supporting Information files will be freely available online, and any third party is permitted to access, download, copy, distribute, and use these materials in any way, even commercially, with proper attribution. For these reasons, we cannot publish previously copyrighted maps or satellite images created using proprietary data, such as Google software (Google Maps, Street View, and Earth). For more information, see our copyright guidelines: http://journals.plos.org/plosone/s/licenses-and-copyright.

Response: We measured the topographic data and processed the topographic map in three dimensions and contour color separation. The graphic is made by us and there is no copyright problem.

Point: 3.2. If you are unable to obtain permission from the original copyright holder to publish these figures under the CC BY 4.0 license or if the copyright holder’s requirements are incompatible with the CC BY 4.0 license, please either i) remove the figure or ii) supply a replacement figure that complies with the CC BY 4.0 license. Please check copyright information on all replacement figures and update the figure caption with source information. If applicable, please specify in the figure caption text when a figure is similar but not identical to the original image and is therefore for illustrative purposes only.

Response: we have measured the topographic data along the line and processed the topographic map in three dimensions and contour color separation. The graphic is made by ourselves and there is no copyright problem.

Point: 4. Please upload a copy of Figure 3, to which you refer in your text on page 10. If the figure is no longer to be included as part of the submission please remove all reference to it within the text.

Response: We had upload a copy of Figure 3.

Response to Reviewer’s Comments

Reviewer #1: 

Point 1: The results of the research should be in the abstract. Materials and Methods should be corrected for example see the following papers:

We revised the reference documents and improved the materials and methods.

Pamucar, D., & Ćirović, G. (2018). Vehicle route selection with an adaptive neuro fuzzy inference system in uncertainty conditions. Decision Making: Applications in Management and Engineering, 1 (1), 13-37.

Liu, F., Aiwu, G., Lukovac, V., & Vukic, M. (2018). A multicriteria model for the selection of the transport service provider: A single valued neutrosophic DEMATEL multicriteria model. Decision Making: Applications in Management and Engineering, 1 (2), 121-130.

Noureddine, M., & Ristic, M. (2019). Route planning for hazardous materials transportation: Multicriteria decision making approach. Decision Making: Applications in Management and Engineering, 2 (1), 66-85.

Blagojević, A., Vesković, S., Kasalica, S., Gojić, A., & Allamani, A. (2020). The application of the fuzzy AHP and DEA for measuring the efficiency of freight transport railway undertakings. Operational Research in Engineering Sciences: Theory and Applications, 3 (2), 1-23.

Kasalica, S., Obradović, M., Blagojević, A., Jeremić, D., & Vuković, M. (2020). Models for ranking railway crossings for safety improvement. Operational Research in Engineering Sciences: Theory and Applications, 3 (3), 84-100.

Response：We revised the References and improved the Materials and Methods.

Point 2: Algorithms cannot be results. Algorithms are a paper methodology. Correct this and better explain the methodology. The selection Discussion and empirical research is debatable. First is the Case study, then the results. There is no discussion. The results are incomplete. One result must be obtained with the TOPSIS method, and then a sensitivity analysis must be performed, in order to exclude some criteria, etc. It is necessary to make scenarios and then discuss all these results.

Response：We have updated the Discussion and empirical research. The sensitivity analysis with variable Rho is added base on the comments. After analysis, the variable Rho directly affects the closeness and project. 

Point 3: The conclusion should include the most important results, research limits and guidelines for future research.

Response：We rewrote the conclusions and added research results, research limitations and guidelines for future research.

Point 4: Adjust the paper according to the guidelines.

Response：Revised according to comments.

Reviewer #2: 

Point 1: I suggest the authors to improve the introduction section. Authors should better highlight the objective of their work and to what extent it contributes to close a gap in the existing literature and/or practice. What is the innovative value of the contribution proposed by the authors?

Response：We rewrote the Introductions, highlight the objectives of previous work and the extent to which it contributes to bridging gaps in existing literature and practice.

Point 2: In introduction section authors should provide more information about existing MCDM models used in the field and their benefits/weaknesses.

Response：Revised according to comments.

Point 3: Why you have used TOPSIS method? Why not RAFSI, MABAC, MAIRCA, VIKOR methdos? This should be discussed. The authors need to discuss their contributions compared to those in related papers. The authors must clearly discuss the significance of the research problem in the first section.

Response：Revised according to comments. It is supplemented in Introduction.

Point 4: Add separate literature review section. You should provide more recent references published in last two-three years. Remove references published before 2017.

Response：Revised according to comments. Several references have been added and removed some references published before 2017.

Point 5: The authors should explain why they insist in practically using the Comprehensive weight model. Why not LBWA, FUCOM or BWM models?

Response：Revised according to comments. It is supplemented in TOPSIS Evaluation Model. LBWA, FUCOM or BWM models can also be used. However, considering that TOPSIS is simpler and can make full use of the existing data, it is more suitable for internal sorting of sample data, this paper adopts TOPSIS.

Point 6: You should provide step by step calculations for provided methodology, especially Comprehensive weight model. You should explain in detail this methodology.

Response：In this paper, Materials and Methods are the whole process of calculation, which is proved in Appendix A. We also added the explanation of preference coefficient according to the Comments.

Point 7: Explain in more details in the data used in the case study, the data for the testing, the criterion for the accuracy, and others to claim these points.

Response：Revised according to comments. It is supplemented in Discussion and empirical research.

Point 8: Validation section is missing. How we can judge about these results? Comparisons with existing algorithms from the literature is missing.

Response：Revised according to comments. The comprehensive weight vector increases the preference coefficient ρ, which is essential different from existing other comprehensive weight methods. T

Point 9: Discussion section is missing. How should we know about the quality of these solutions? Could you compare these results with some existing approaches in literature? The improvement must be discussed.

Response：Revised according to comments. The comprehensive weight vector increases the preference coefficient ρ, which is essential different from existing other comprehensive weight methods. The preference coefficient ρ which can be determined by the accuracy of subjective and objective information and preference. When the decision-maker obtains the subjective weight and objective weight, through comprehensive analysis of the project, a preference coefficient from the perspective of the interests of the decision-maker is proposed, which can better serve the decision-maker. Intelligent computing can be used to form an evaluation of the scheme by a large number of evaluation groups (including experts, stakeholders, etc.), and the evaluation results are more objective. 

Point 10: The conclusion section seems to rush to the end. The authors will have to demonstrate the impact and insights of the research. The authors need to clearly provide several solid future research directions. Clearly state your unique research contributions in the conclusion section. Add limitations of the model. No bullets should be used in your conclusion section.

Response：Revised according to comments. 

Reviewer #3:

Point: 1.It is suggested that the authors introduce the organization of this paper in the end of introduction.

Response：Revised according to comments. 

Point: 2. Page 9, line 5, “and each scheme m has a quantitative value B_{ij}” should be “and each scheme has a quantitative value b_{ij} for each quantitative index”.

Response: We have modified for “and each scheme has a quantitative value Bij for each quantitative index” ( Line 24 ).

Point: 3. Page 10, line 16, “and f_i^k is the importance of indicator F_i^k compared with indicator F_j^k” should be “and f_{ij}^k is the importance of indicator F_i^k compared with indicator F_j^k”.

Response: We have modified for “and f_{ij}^k is the importance of indicator F_i^k compared with indicator F_j^k”.” ( Line 24 ).

Point: 4. Page 11, formula (7), what does the symbols “x_{ij}, max(x_j), min(x_j)” stand for? Maybe they are “d_{ij}”, “max_j(d_{ij})” and “min_j(d_{ij})”.

Response : We have modified, “xij, max(xj), min(xj)” are “dij”, “max(dj)” and “min(dj)”

Point: 5. Page 12, formula (14), what does the symbols “f_i” and “z_{ij}” stand for?

Response : fi is comprehensive weighted score, is the evaluation matrix after standardization.

Point: 6. Page 12, formula (15), “where \\rho reflects the decision-maker’s preference for objective or subjective weight.” Does it reflect the preference of decision makers for objective or subjective weights here? The higher the value of \\rho is, does the decision maker prefer subjective weight or objective weight?“where\\rho

Response : where reflects the decision-maker’s preference for subjective weight. The higher the value of ρ is the decision maker prefer subjective weight.

Point: 7. Page 12, in theorem 1, “formula (10)” should be “formula (15)”.

Response : We have modified “formula (10)” for “formula (15)”. Also we have modified “From formula (12)” for “From formula (27)”. 

Point: 8. Page 13, in step 3, “formula (18)” should be “formula (16)”.

Response : We have modified “formula (18)” for “formula (16)”. 

Point: 9. It is suggested that the authors compare the proposed comprehensive weight method with existing comprehensive weight methods.

Response：Revised according to comments. 

Point: 10. It is suggested that the authors add a sensitivity analysis with variable \\rho.

Response：The sensitivity analysis with variable Rho is added according to the comments. After analysis, the variable Rho directly affects the closeness and project selection.

---

## [Decision Letter · Decision Letter 1]

30 Dec 2021

An intelligent optimization method for highway route selection based on comprehensive weight and TOPSIS

PONE-D-21-10954R1

Dear Dr. Liu,

We’re pleased to inform you that your manuscript has been judged scientifically suitable for publication and will be formally accepted for publication once it meets all outstanding technical requirements.

Kind regards,

Dragan Pamucar

Academic Editor

PLOS ONE

Additional Editor Comments (optional):

Reviewers' comments:

Reviewer's Responses to Questions

**Comments to the Author**

1. If the authors have adequately addressed your comments raised in a previous round of review and you feel that this manuscript is now acceptable for publication, you may indicate that here to bypass the “Comments to the Author” section, enter your conflict of interest statement in the “Confidential to Editor” section, and submit your "Accept" recommendation.

Reviewer #1: All comments have been addressed

Reviewer #2: All comments have been addressed

Reviewer #3: (No Response)

2. Is the manuscript technically sound, and do the data support the conclusions?

Reviewer #1: Yes

Reviewer #2: Yes

Reviewer #3: Yes

3. Has the statistical analysis been performed appropriately and rigorously? 

Reviewer #1: Yes

Reviewer #2: Yes

Reviewer #3: Yes

4. Have the authors made all data underlying the findings in their manuscript fully available?

Reviewer #1: Yes

Reviewer #2: Yes

Reviewer #3: Yes

5. Is the manuscript presented in an intelligible fashion and written in standard English?

Reviewer #1: Yes

Reviewer #2: Yes

Reviewer #3: Yes

6. Review Comments to the Author

Reviewer #1: Greetings,

The authors followed all the suggestions. The paper should now be accepted for publication.

All best

Reviewer #2: The authors have addressed the point of my concern. I am happy with their corrections. Hence, I would like to recommend this manuscript to be published.

Reviewer #3: The authors address the issues of the previous version well. But there are still some minor problems.

1. In formula (7), “max(d_j)” ”min(d_j)”should be “man_i(d_{ij})”and “min_i(d_{ij})”. “d_j” is defined in formula (10).

2. In formula (13), “sum_{j=1}^{m}w_l=1” should be “sum_{j=1}^{m}w_j=1”.

3. In formulas (23) and (24), “f_{ij}” should be “z_{ij}”, “f_j^+” should be “z_j^+”, f_j^-” should be “z_j^-”.

7. PLOS authors have the option to publish the peer review history of their article (what does this mean?). If published, this will include your full peer review and any attached files.

Reviewer #1: No

Reviewer #2: No

Reviewer #3: No

---

## [Editor Report · Acceptance letter]

16 Feb 2022

PONE-D-21-10954R1 

An intelligent optimization method for highway route selection based on comprehensive weight and TOPSIS 

Dear Dr. Liu:

I'm pleased to inform you that your manuscript has been deemed suitable for publication in PLOS ONE. Congratulations! Your manuscript is now with our production department. 

Kind regards, 

on behalf of

Dr. Dragan Pamucar 

Academic Editor

PLOS ONE